# Estimation of the Value of Ecosystem Carbon Sequestration Services under Different Scenarios in the Central China (the Qinling-Daba Mountain Area)

**Yuyang Yu [1], Jing Li [1,\*], Zixiang Zhou [2], Li Zeng [1] and Cheng Zhang [1]**

[1] School of Geography and Tourism, Shaanxi Normal University, Xi'an 710119, China; yuyuyang@snnu.edu.cn (Y.Y.); zengli@snnu.edu.cn (L.Z.); zhangcheng@snnu.edu.cn (C.Z.)

[2] College of Geomatics, Xi'an University of Science and Technology, Xi'an 710054, China; zhouzixiang@snnu.edu.cn

\* Correspondence: lijing@snnu.edu.cn

**Abstract:** The Qinling-Daba Mountain area is a transitional zone between north and south China and not much is known about its carbon storage, particularly its pool of soil organic carbon (SOC). Given this shortcoming, more reliable information regarding its SOC is needed. In light of this, we quantified above and below-ground carbon sinks using both the Carnegie-Ames-Stanford approach (CASA) model and an improved carbon cycle process model. We also assessed the net present value (NPV) for carbon budgets under different carbon price and discount rate scenarios using the NPV model. Our results indicated that the net primary productivity (NPP) was lower in places with low density forests that were situated at high elevation. The spatial distribution of carbon storage depended on NPP production and litter decompositon, which reflected specific vegetation as well as temperature and moisture gradients. The lowest amounts of carbon storage were in the center of the Qinling Mountains and also partly in the Daba area, which is a location associated with sparse grassland. Contrastingly, the broad-leaved forested area showed the highest amount of carbon storage. NPV was positively correlated with discount rate and carbon prices, thus resulting in the highest values in the forests and grassland. The net present value of total soil carbon sequestration in the six scenarios in 2015 was 3.555 b yuan, 3.621 b yuan, 5.421 b yuan, 5.579 b yuan, 7.530 b yuan, 7.929 b yuan; The net present value of total soil carbon sequestration in 6 scenarios in 2017 is 2.816 b yuan, 2.845 b yuan, 4.361 b yuan, 4.468 b yuan, 6.144 b yuan, 6.338 b yuan (billion = $10^9$; b; RMB is the legal currency of the China, and its unit is yuan, 1 euro = 7.7949 yuan, and 1 pound = 9.2590 yuan). Levying a carbon tax would be a notable option for decision makers as they develop carbon emission reduction policies. Given this, incorporating discount rates and carbon pricing would allow for more realistic value estimations of soil organic carbon. This approach would also provide a theoretical basis and underscore the practical significance for the government to set a reasonable carbon price.

**Keywords:** net primary productivity; soil organic carbon; soil respiration; net present value; China's north-south transition zone

---

## 1. Introduction

Increasing economic development coupled with accelerated urbanization has led to a rapid increase in anthropogenic carbon dioxide emissions. This has resulted in increased carbon dioxide concentration in the atmosphere [1], which has led to rising global temperatures [2]. One important option in alleviating the many social problems caused by this temperature rise is to increasing the carbon sink capacity of terrestrial ecosystems [3]. Soil is the basis of human survival, providing food, nutrition and various resources (including carbon, primary minerals, secondary minerals, organic

matter and microorganisms), and also an important part of the global carbon cycle. Soil organic carbon (SOC) decomposition also releases carbon dioxide, which is an important, naturogenic cause of elevated atmospheric carbon dioxide concentration that also has a direct impact on global warming. SOC is important in the global carbon cycle and an integral part of many initiatives and policies to mitigate climate change [4]. To better account for these emissions, carbon accounting has become the most important task for the National Development and Reform Commission to deploy the China unified carbon market construction. Such accounting approaches are urgently needed to allow for the calculation of soil carbon storage and the estimate of its value.

Geostatistical methods [5–7] are commonly used to study the spatial variability of SOC, and recently some machine learning methods frequently used to predict SOC [8–10]. Many non-linear models including Cubist (Cu), Random Forest (RF), Regression Tree (RT) and a Multiple Linear Regression (MLR) were used to simulate the distribution of soil carbon reserves, in order to predict and generate a continuous spatial clear soil carbon map [8–10]. In the field of soil monitoring, the strict geostatistical requirements for sampling and subsequent laboratory analysis limit its practicability; to this end, remote-sensing technology has obvious advantages [11]. The application of this method in China has primarily focused on the estimation and spatial distribution of SOC pools at a national scale [12], the impact of wind erosion on soil carbon pools [13], and the impact mechanism of SOC pools at a regional scale [14,15] as well as SOC changes across a time series [16]. Many methods have been previously applied to SOC estimation, including direct estimation of remote-sensing imaging, indirect estimation of vegetation index, and non-imaging spectrometry estimation [17]. Generally speaking, there are some problems in the application of remote sensing estimation, field investigation and statistical data modeling in organic carbon estimation. With the improvement of remote sensing inversion technology, the estimation method of organic carbon is more accurate. Therefore, this paper combines the carbon cycle process model of an ecosystem with the estimation model of net primary productivity to make the simulation results more ideal.

Commonly used methods to calculate domestic carbon sequestration estimates include afforestation cost, industrial oxygen production, carbon tax, and market value. In 2011, Li and Ren [18] estimated the distribution characteristics and changes of carbon fixation value and oxygen release value in the loess plateau of northern Shaanxi, and adopted two methods: afforestation cost and industrial oxygen production. In 2014, Kong and Zhang [19] estimated the functional value of carbon sequestration of wetlands in protected areas by using the carbon tax law, providing a certain reference for the functional value of carbon sequestration of wetlands in China. In 2014, Pang et al. [20] used the market value method to quantitatively evaluate the ecosystem service value provided by the Zoige alpine wetlands. The net present value of a project is the present value of current and future benefits minus the present value of current and future costs. [21]. The net present value mainly measures the profit index of economic evaluation; because its theory foundation is relatively perfect and has important economic significance, it is widely applied in each domain. For example, the NPV method has been used to estimate the economic value associated with air pollution, soil fertility management, greenhouse gas emission reductions, and other ecological projects such as agricultural land value estimates [22–24]. Albornoz et al. [25] uses the NPV method to judge whether a project has investment criteria or not. Elariane et al. [26] examine the feasibility of such projects by assessing the NPV of the smart and economic housing model. In 2014, Baral et al. [27] estimated NPV to assess and compare the economic value between ecosystem services in different scenarios; their study found that the higher the discount rate, the greater the value of the corresponding value. In 2015, Tremblay et al. [28] applied NPV to small-scale agricultural income accounting in the Brazilian Amazon. Their study concluded that there was a significant difference between different discount rates and agricultural income. Nghiem et al. [29] used NPV method to determine how to estimate the conservation benefits of carbon sequestration and biodiversity under different forest management models to maximize the NPV of timber sales. Previous studies have found that NPV method is applicable to various fields and has certain advantages in value estimation. This paper combines the NPV method with the application

of soil organic carbon in geography. It is found that the NPV method, which incorporates carbon price and discount rate into the soil carbon sequestration model, is a more practical method compared with the afforestation cost method, carbon tax law method and cost avoidance method.

China's carbon trading market is still in the pilot stage, and the discount rate has certain directivity to the economy. The NPV method is an economic method combined with carbon tax and discount rate, while the soil organic carbon estimation method combined with ecosystem carbon cycle process model and net primary productivity estimation model is now a more accurate evaluation method, combined with the economic method and remote sensing method, this is a new attempt to evaluate the value of soil organic carbon. Given this, we sought to combine soil organic carbon with a carbon cycle process model to estimate the spatial distribution of soil respiration using a CASA (Carnegie-Ames-Stanford approach) model. We also established a linear regression model for soil organic carbon storage and soil respiration, and then estimated soil organic carbon (30 cm) reserves. Based on this approach, we then used the NPV method to estimate soil organic carbon value in the Qinling-Daba Mountain. This study regarding carbon value according to a scenario mode will provide a useful reference for the future developments in the government's macro-control of carbon prices.

## 2. Study Area and Data

### 2.1. Study Area

China's north-south transition zone-known as the Qinling-Daba Mountain is located in the western-central region of China. The complete area is located between 102°54′ to 112°40′ E and 30°50′ to 34°59′ N, covering the provinces of Shaanxi, Gansu, Sichuan, Hubei, Henan, and Chongqing city [30,31] (Figure 1). The Qinling-Daba Mountain is the upstream part of the Hanjiang River, which area of the basin in this area reaches 62,000 km$^2$ [32]. The northern branch of the Hanjiang River basin originates from the southern slope of the Qinling Mountains and the southern branch from the northern slope of the Daba Mountains. The northern slope of Qinling Mountains has steep terrain, many canyons, and the southern foothills are long and gentle. The northern foot of the Qinling Mountains is a large fault cliff. The northern slope is short and steep, forming many gorges. The southern slope is long and gentle, with many mountains and intermountain basins. The Daba Mountains are composed of hard crystalline limestone, with a high mountain range and a well-developed Lysite landform. There are many large-scale karst depressions, caves, funnels and karst springs [33].

The region is located in the transition zone between the warm temperate zone and the north subtropical zone, and the vegetation floristic composition is complex and diverse, mainly reflected in the vegetation type dominated by the deciduous broad-leaved forest in the main body of the Qinling mountains, and the mixed vegetation type of deciduous broad-leaved forest containing evergreen broad-leaved forest in the south of the Qinling mountains [34,35], detailed information on vegetation classification is shown in Table 1. The region has a large elevation drop, steep mountain topography, less man-made destructive activities, and mild climate, which makes natural vegetation flourish, rich soil resources and various types, mainly including brown earths, yellow-brown earths, skeletal soils, litho soils, paddy soils, cinnamon soils, yellow-cinnamon soils and so on [36], the classification system of soil is shown in Table 2.

### 2.2. Data Sources

The data sources used in this study were as follows: remote-sensing data, including 250 m spatial resolution MOD13Q1 data obtained from 2003–2017, (http://data.cma.cn/site/index.html); the Harmonized World Soil Database (HWSD), which provided 30 cm, soil surface organic carbon content for 1:1,000,000 soils in China; soil and vegetation data mainly comes from the Resource and Environment Data Cloud Platform of the Chinese Academy of Sciences (http://www.resdc.cn/DataSearch.aspx); China's meteorological science data sharing service network, which included data on monthly average precipitation, monthly average temperature, and surface solar radiation.

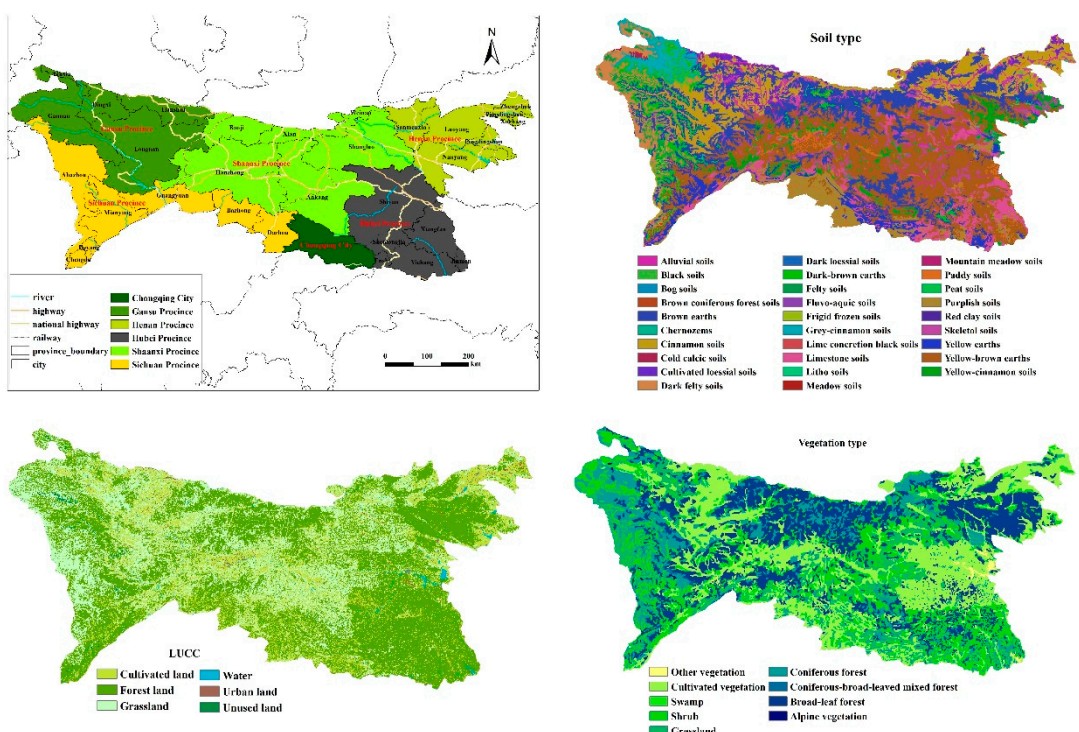

**Figure 1.** Location of the study area.

**Table 1.** Information about the vegetation type data source (the definition of vegetation type comes from the Resource and Environment Data Center of Chinese Academy of Sciences-Resource and Environment Data Cloud Platform http://www.resdc.cn/DataSearch.aspx).

| Vegetation Type | Definition |
|---|---|
| Coniferous forest | Coniferous forest is a general term for all kinds of forests composed of coniferous trees for the construction of population. |
| Broad-leaf forest | Broad-leaf forest is composed of broad-leaved trees, with broad leaves, compared with coniferous forest and common leaves. |
| Coniferous-broad-leaved mixed forest | Coniferous broad-leaved mixed forest refers to the transition type between cold temperate coniferous forest and summer green broad-leaved forest. It is usually composed of broad-leaved tree species such as Quercus, Acer, Tilia and some species of Picea, Abies and Pinus. |
| Shrub | Shrub refers to the type of vegetation dominated by shrubs. Most of the species are medium-growing and clustered shrub life forms. The community height is usually less than 5 m, and the canopy density is usually 0.3–0.4. Thickets are often found in places where the climate is too dry or cold and forests are difficult to grow. |
| Grassland | Grassland is one of the earth's ecosystems and is the most widely distributed vegetation type on the earth. Meadow refers to the vegetation type with perennial mesophytic herbs as the main body developed under moderate water conditions. Grass refers to herbaceous plant communities, including grasses and non-grasses. |
| Swamp | Swamp refer to the surface and subsurface soils that are often over-wetted, with wet plants and marsh plants growing on the surface. |
| Alpine vegetation | Alpine vegetation refers to a variety of community types consisting of cold-resistant, drought-tolerant and snow-tolerant plants located in the alpine zone. |
| Cultivated vegetation | Cultivated vegetation refers to wild plants which have certain production value or economic characteristics, stable heredity and can meet human needs after artificial cultivation. |
| Other vegetation | Other vegetation types include water area, unused land, wasteland, etc. |

## 2.3. Models

According to relevant models, we estimated the carbon sequestration above and belowground, and set up different scenarios to evaluate the carbon sequestration according to different indicators. The method scheme is shown in Figure 2.

**Table 2.** Classification system of soil types (the Classification system of soil types comes from the Resource and Environment Data Center of Chinese Academy of Sciences-Resource and Environment Data Cloud Platform http://www.resdc.cn/DataSearch.aspx).

| Soil Order | Great Soil Group |
|---|---|
| Leached soil | Brown coniferous forest soils |
| | Brown earths |
| | Yellow-brown earths |
| | Yellow-cinnamon soils |
| | Brown earths |
| | Dark-brown earths |
| | Albic soils |
| Semi-leaching soil | Torrid red soils |
| | Cinnamon soils |
| | Grey-cinnamon soils |
| | Black soils |
| | Grey forest soils |
| Calcium layer soil | Chernozems |
| | Castanozems |
| | Castano-cinnamon soils |
| | Dark loessial soils |
| Arid soil | Brown pedocals |
| | Sierozems |
| Desert soil | Gray desery soils |
| | Gray-brown desrt soils |
| | Brown desert soils |
| Primary soil | Cultivated loessial soils |
| | Red clay soils |
| | Alluvial soils |
| | Takyr soils |
| | Aeolian soils |
| | Limestone soils |
| | Volcanic soils |
| | Purplish soils |
| | Litho soils |
| | Skeletol soils |
| Semi-aqueous soil | Meadow soils |
| | Lime concretion black soils |
| | Mountain meadow soils |
| | Shruby meadow soils |
| | Fluvo-aquic soils |
| Aqueous soil | Bog soils |
| | Peat soils |
| Saline soil | Saline soils |
| | Desert solonchaks |
| | Coastal solonchaks |
| | Acid sulphate soils |
| | Frigid plateau solonchaks |
| | Solonetzs |
| Artificial soil | Paddy soils |
| | Cumulated irrigated soils |
| | Irrigated desert soils |
| High-Mountain soils | Felty soils |
| | Dark felty soils |
| | Frigid calcic soils |
| | Cold calcic soils |
| | Cold brown calcic soils |
| | Frigid desert soils |
| | Cold desert soils |
| | Frigid frozen soils |
| Iron bauxite | Humid-thermo ferralitic |
| | Lateritic red earths |
| | Red earths |
| | Yellow earths |

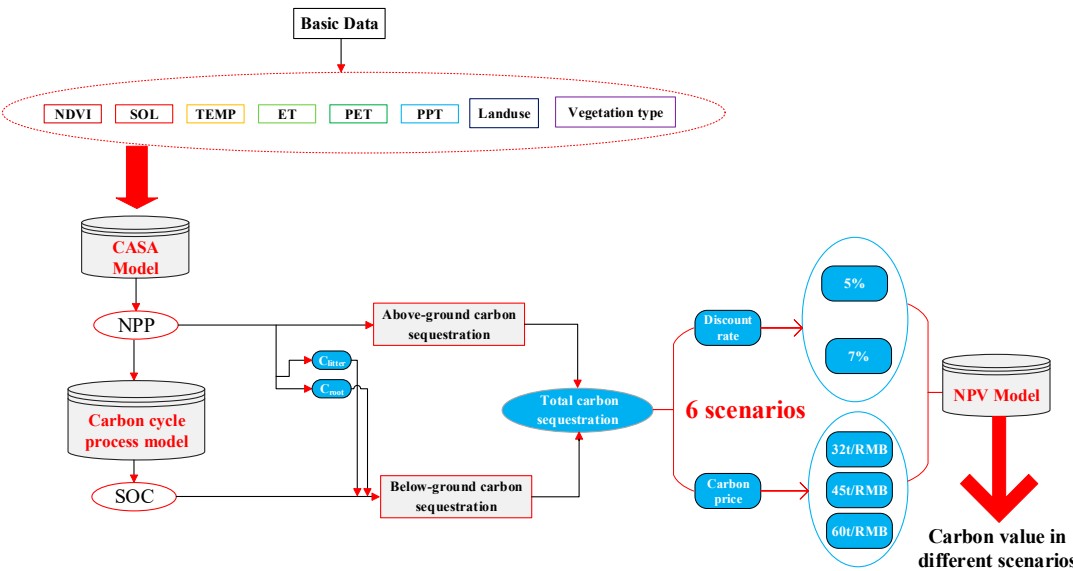

**Figure 2.** Framework of study.

### 2.3.1. Aboveground Carbon Sequestration

NPP refers to the organic mass accumulated by green plants in unit time and unit area. It is the remaining part of the total organic matter produced by plants after photosynthesis minus autotrophic respiration. CASA [37,38] mainly extracted normalized vegetation index (NDVI) from remote sensing images. Based on the principle of light energy utilization, the model is used to estimate the NPP of global terrestrial ecosystem. The model mainly calculates the NPP of the vegetation using two parameters: the absorbed photosynthetic active radiation (APAR) and the light energy conversion rate $\varepsilon$ [37,38]. The equation is as follows:

$$NPP(x,t) = APAR(x,t) \times \varepsilon(x,t) \tag{1}$$

$$C_a(x,t) = NPP(x,t) \times \sigma \tag{2}$$

The parameters $APAR(x,t)$ (MJ m$^{-2}$ month$^{-1}$) represents the amount of photosynthetic active radiation absorbed by element $x$ in month $t$; $\varepsilon(x,t)$ is a factor that reflects the efficiency with which light energy is used to produce organic compounds in grid $x$ for month $t$; $NPP(x,t)$ (gC MJ$^{-1}$) represents the amount of net primary production aboveground; $C_a(x,t)$ represents the carbon aboveground, and $\sigma$ is the carbon content of organic matter, $\sigma = 0.4$ [39].

### 2.3.2. Below-Ground Carbon Sequestration

Below-ground carbon consisted of three components: SOC, carbon in litter, and carbon in biological roots. For the aboveground carbon cycle, the estimation of NPP was based on photosynthesis process. For underground biomass, we considered the litter and roots of the vegetation. As previously described in the literature [39–42], we used the ratio of below-above-ground biomass (Table 3) to estimate the underground biomass carbon sequestration of different communities (i.e., cultivated land, forests, and grasslands). Remote-sensing images described the differences between cultivated land, forest, and grassland. This approach accounted for the differences between litter and root biomass in different communities [40–42].

$$C_u = SOC + C_{litter} + C_{root} \tag{3}$$

$$T = C_u + C_a \tag{4}$$

**Table 3.** Relationship between carbon in root/litter and above-ground carbon relative to different land types [39–42].

| Specific Relation | Cultivated Land | Forest | Grassland |
|---|---|---|---|
| Root/aboveground | 18.85% | 28.30% | 169% |
| Litter/aboveground | 45.60% | 14.39% | 25.30% |

The parameters $C_u$, $SOC$, $C_{litter}$, and $C_{root}$ describe the amount of below-ground carbon, SOC, litter carbon, and root carbon, respectively. Past studies have established the theoretical relationship between the components of dead branches and fallen leaves comprising NPP and soil organic carbon reserves [43–45]; however, the long-term impact of climate change on carbon emissions remains unclear [46]. $T$, $C_a$, and $C_u$ represent the total carbon sequestration, above-ground carbon, and below-ground carbon, respectively.

Owing to limited data sources, the SOC was calculated indirectly. Soil respiration (SR) is an essential variable for measuring the rate at which soil releases carbon dioxide [37]. Previous studies have confirmed a significant negative correlation between soil respiration and SOC [47]; therefore, SOC can be quantified using SR and regression models.

SR refers to a combination of factors, including the plant roots in the soil, food debris (e.g., animals, fungi, and bacteria metabolic activity), consumption of organic matter, and the process of generating carbon dioxide. This activity is often called heterotrophic respiration, and is part of foundational SR. Zhou et al. [48] published an approach to improve the carbon cycle process model to invert soil respiration, build the relationship between SOC and $CO_2$ emissions in the soil, this model considered moisture factor, using annual precipitation and a factor accounting for annual potential evaporation temperature sensitivity to describe the influence of soil moisture on soil respiration. This approach was more consistent with the actual, on the ground conditions in the study area. The formula used is as follows:

$$A_{ij} = \frac{NPP}{exp(b \times T) \times y}$$
$$y = \frac{1}{1 + 30.0 \times exp(-8.5 \times x)} \tag{5}$$
$$x = \frac{PPT}{PET}$$

$$SOC = f(A_{ij}) \tag{6}$$

where parameters $A_{ij}$ is the soil respiration, refers to the amount of soil respiration at 0 °C without water stress [49]; $PPT$ is the annual precipitation; $PET$ is the annual potential evapotranspiration; $b$ is the temperature sensitivity constant; $x$ is the ratio of precipitation to evapotranspiration, and $y$ is the limit of water on soil respiration, which is between [0.03,1].

Based on the combination of NPP data from the CASA model and carbon cycle process model, the study calculated the annual basic soil respiration. Based on this, a total of 500 sampling points were randomly selected and the inversion data of 500 random points for soil respiration and data from the second National Soil Census were assessed. A linear regression was conducted to compare the two values (Figure 3). The determination coefficient $R^2$ between the two is 0.72, and the results passed the 0.001 significance test. (unit: $t \, hm^{-2} \, a^{-1}$ stands for ton/square hectare per year).

2.3.3. Estimation Model of Net Present Value (NPV)

This study adopted the NPV model established by Polglase to incorporate the discount rate into the estimation model [50], making the calculation results more practical. The calculation formula is as follows:

$$NPV_{ijs} = PVB_{ij} - PVC_{js} \tag{7}$$

where parameters $NPV_{ijs}$ represents the current value of revenue, which is obtained based on the carbon price; $PVB_{ij}$ represents the value of carbon sequestration; and $PVC_{js}$ represents the cost of each price scenario.

$$PVB_{ij} = \sum_{t=0}^{T} \frac{p_i \times q_{tj}}{(1+r)^t} \qquad (8)$$

where parameters $p_i$ represents the price of carbon; $q_{ti}$ represents the annual carbon storage capacity; $r$ represents the annual discount rate; and $t$ represents the time cycle. The carbon price refers to the price of China's carbon trading market and the carbon price survey results (2013) [51,52] and the discount rate was set according to China's national conditions and based on the discount rate value used to calculate forest value with respect to some European countries [22]. This study used 5% and 7% discount rates as scenario parameters.

$$PVC_{js} = EC_j + \sum_{t=0}^{T} \frac{MC \times PFE_s}{(1+r)^t} \qquad (9)$$

The parameters *PFEs* represents the opportunity cost of planting and afforestation carbon; $EC_j$ is the one-time cost; and *MC* is the annual carbon maintenance and transaction cost.

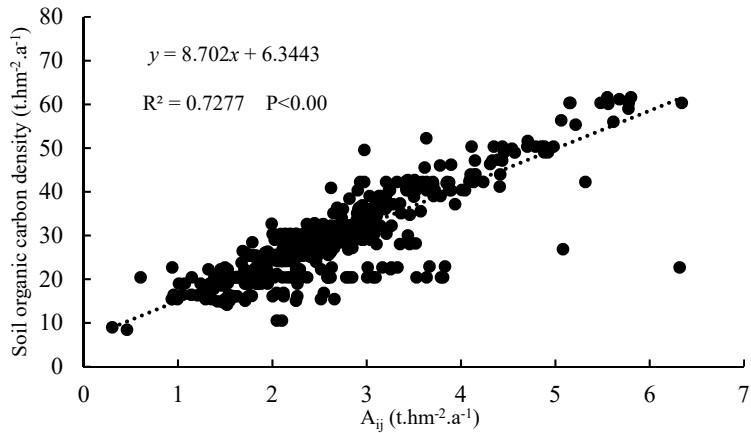

**Figure 3.** Correlation analysis between soil respiration and soil organic carbon.

## 3. Result

### 3.1. Spatiotemporal Analysis

Based on the remote sensing data, the CASA model was used to estimate the mass of NPP across 2003–2017. The annual average NPP of different vegetation types showed interannual differences (Table 4). The decreasing order of mean value was as follows: broad-leaved forest, cultivated vegetation, grassland, shrub, swamp, coniferous-broad-leaved mixed forest, coniferous forest, alpine vegetation, and other vegetation.

In the examine area (Figure 4), the distribution pattern was "middle high, low ambient". The high values were mainly concentrated in the Qinling and Daba Mountain, where had higher altitude and higher NDVI. NPP and NDVI were positively correlated. In addition to the high altitude, there was a low population density, with a correspondingly lower ability to decrease vegetation cover in the mountainous. At the same time, the extensive implementation of the policy of returning farmland to forests and grassland will increase the carbon sequestration capacity of forests and grasses in the region to a certain extent. Low values were primarily in Huining and Yuzhong counties and other northern regions in the southern Gansu province.

**Table 4.** Annual average NPP of different vegetation types in 2003–2017, unit: gC per square meter per year.

| Vegetation Types | 2003 | 2005 | 2010 | 2015 | 2017 |
|---|---|---|---|---|---|
| Coniferous forest | 442 | 466.1 | 452.8 | 564 | 468 |
| Coniferous-broad-leaved mixed forest | 509 | 518.3 | 500 | 634.6 | 520.7 |
| Broad-leaf forest | 917 | 977.8 | 939.6 | 1155 | 992.8 |
| Shrub | 573.9 | 672.9 | 545 | 562 | 521.3 |
| Grassland | 580.9 | 625.5 | 623.3 | 776.8 | 643.8 |
| Swamp | 505.3 | 523.7 | 544.6 | 673.2 | 455.8 |
| Alpine vegetation | 428 | 423.3 | 431.1 | 528.6 | 419.6 |
| Cultivated vegetation | 592.7 | 642.2 | 636.6 | 764.4 | 668.7 |
| Other vegetation | 387.3 | 432.3 | 435 | 485.9 | 429.6 |

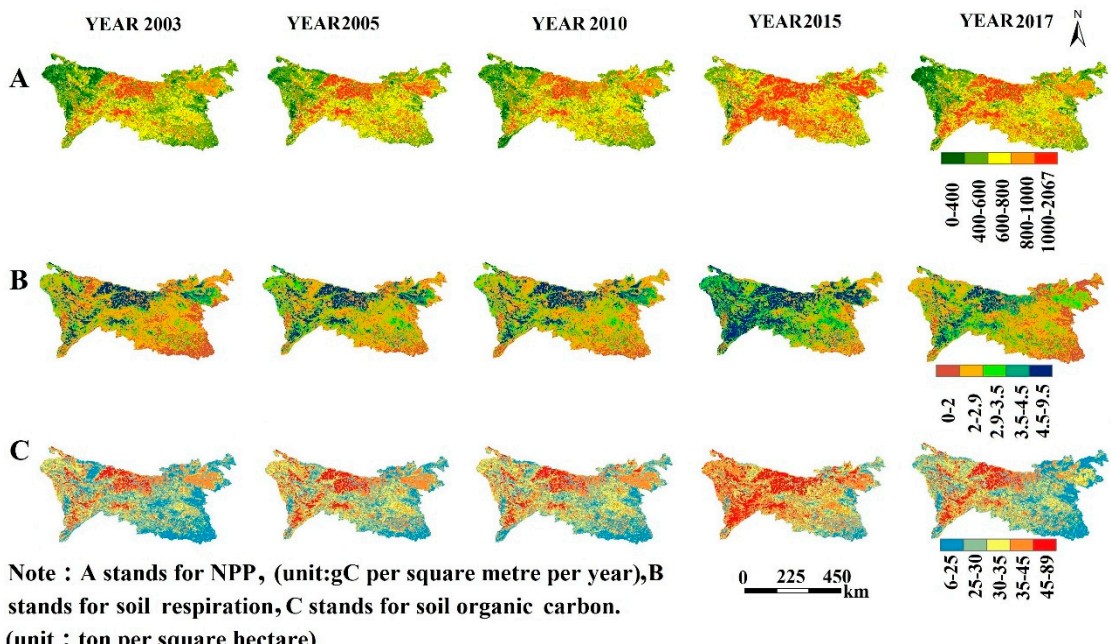

Note：A stands for NPP，(unit:gC per square metre per year),B stands for soil respiration,C stands for soil organic carbon.
(unit：ton per square hectare)

**Figure 4.** Spatial and temporal distribution of NPP, soil respiration, soil organic carbon in 2003–2017.

## 3.2. Spatial Distribution of Soil Organic Carbon

The annual spatial and temporal distribution of soil basic respiration was obtained by combining the NPP data obtained from the CASA model with the carbon cycle process model. From Figure 4, the soil respiration had a lower distribution towards the northwest and southeast regions. As the western area is largely mountainous with high altitudes, the main factor behind this pattern was temperature; resulting in significantly lower temperatures relative to the eastern area. The soil foundation of respiration inversion was conducted on the basis of the temperature, leading to the east to west pattern of lower to higher temperatures, respectively.

Using a simulated linear regression model and soil respiration data, the SOC distribution map covering 250 m was inversely presented. The figure shows that the spatial distribution of soil organic carbon is consistent with that of soil respiration where is higher in the west than the east; the high value in the west mainly appeared in the north of the Qinling and Daba Mountains. The NPP and SOC in the central region are higher than in the eastern region. Because that the central region has good hydrothermal conditions; this leads to the region having high vegetation coverage and primary productivity, and jointly improves the regional SOC.

In terms of interannual change, the year 2003–2015 continued to rise, and the year 2015–2017 was in a downward trend. Climate factors were the main indicators of NPP in restricted areas. Precipitation

and temperature in 2015 were higher than those in other years, evapotranspiration was larger, and solar radiation was strong, which was conducive to vegetation growth, vegetation growth was good, and in a certain process. The carbon storage in 2015 is higher than that in other years because of the promotion of carbon storage and the positive effect of human activities on vegetation growth. However, the carbon storage in 2015–2017 is in a downward trend. The main reason is that the economic development has led to the destruction of better vegetation areas such as forests and grasslands, which has led to the construction of forests and grasslands. The increase of land use, to a certain extent, affects the local climate, making the local temperature rise, evaporation changes, precipitation reduction and other negative effects, and ultimately will have a negative impact on the local ecological environment, making carbon storage decline.

*3.3. Total Carbon Sequestration*

Figure 5 shows that from 2003 to 2017, total carbon sequestration fluctuates in interannual changes, mainly due to the impact of national policies on vegetation restoration, coupled with the change of land use patterns. The figure shows that underground carbon sequestration is higher than aboveground carbon storage, mainly because underground carbon storage is directly related to the machinery of soil respiration. Biomass is mainly the aboveground part, which is related to NPP. There are abundant vegetation types and high forest coverage in Qinling-Daba Mountain, where is many national forest parks, which are well protected by forests. At the same time, ecological projects such as returning farmland to forestry and grassland and afforestation have been carried out in many areas of Qinling-Daba Mountain, which has promoted the promotion of forest and herbaceous biomass in the study area. Soil carbon pool is mainly underground, which is related to soil respiration. Vegetation fixes atmospheric $CO_2$ through photosynthesis and forms organic carbon of plants, then enters the soil in the form of litter. Some soil organic carbon returns to the atmosphere in the form of $CO_2$ through the mineralization process of litter, while the other part of organic carbon returns to the atmosphere in the form of $CO_2$. Carbon is transformed into soil humus through humification. Soil humus is further decomposed by microorganisms to release $CO_2$ to the atmosphere. Under the influence of climate factors, soil organic carbon gradually reaches dynamic equilibrium during the long-term evolution of soil. At this time, the organic carbon released through litter into soil is equal to that released through soil respiration. Soil organic carbon as the raw material of soil respiration, directly affects the soil respiration intensity.

The content of soil organic carbon depends on the input and output of soil organic carbon. That is, soil organic carbon content is determined by net primary productivity of vegetation and soil organic carbon mineralization intensity. Mineralization intensity is mainly affected by climatic conditions (precipitation, temperature, radiation, etc.) [53]. The soil organic carbon in the middle of the Qinling-Daba Mountain was higher than that around the mountain, this was the result of the high vegetation density and primary productivity resulting from the good temperatures and precipitation conditions in these areas Soil organic carbon in southwestern Qinling-Daba Mountain was also relatively high, this was due to a high net primary productivity, giving rise to a large amount of litter input every year. Coupled with the low temperature, the decomposition rate of wood litter was also low. However, in the southern part of the region, soil organic carbon was low, mainly because the warm climate promoted the decomposition of soil organic carbon. This resulted in low soil organic carbon content in the region, which was consistent with the relatively low soil respiration in the region.

Based on the time scales used to explore the vegetation types (Figure 6), we analyzed the effect of different vegetation types on the carbon layers. As time passed, different vegetation types increased in the carbon layers. When conducting an analysis from the perspective of vegetation type, the broad-leaved forest had the most abundant carbon reserves. The order of vegetation according to decreasing carbon reserves was as follows: broad-leaved forest, grassland, cultivated vegetation, shrub, coniferous broad-leaved mixed forest, swamp, coniferous forest, alpine vegetation, and other vegetation.

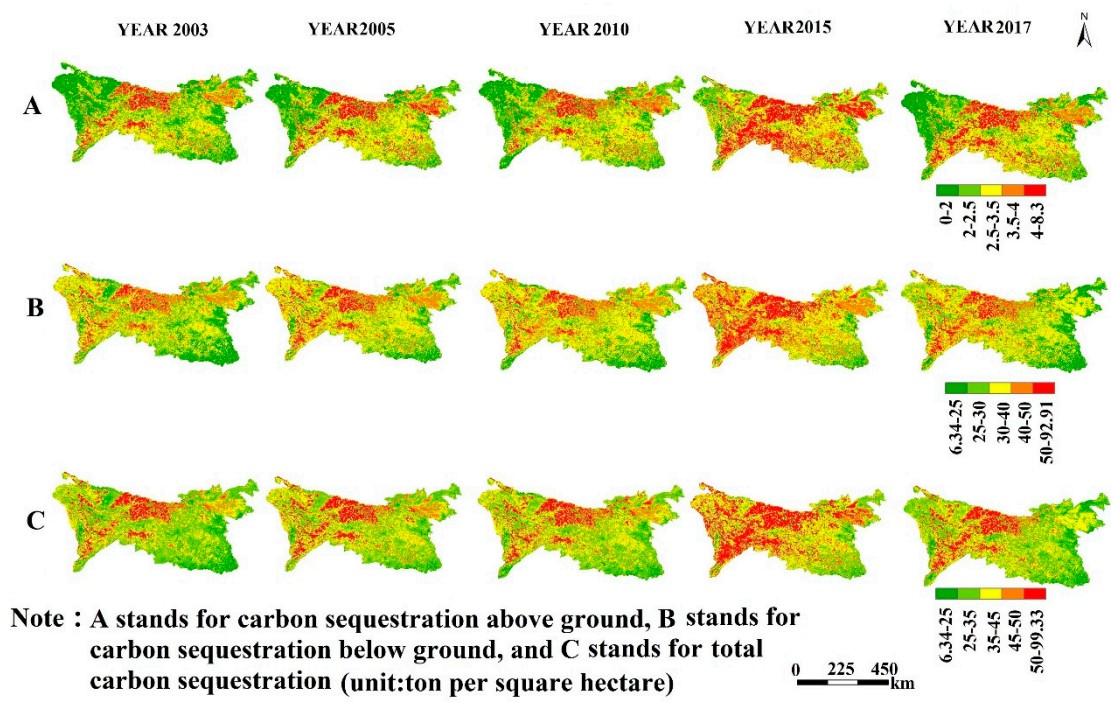

**Figure 5.** Spatial distribution of total carbon sequestration in the study area.

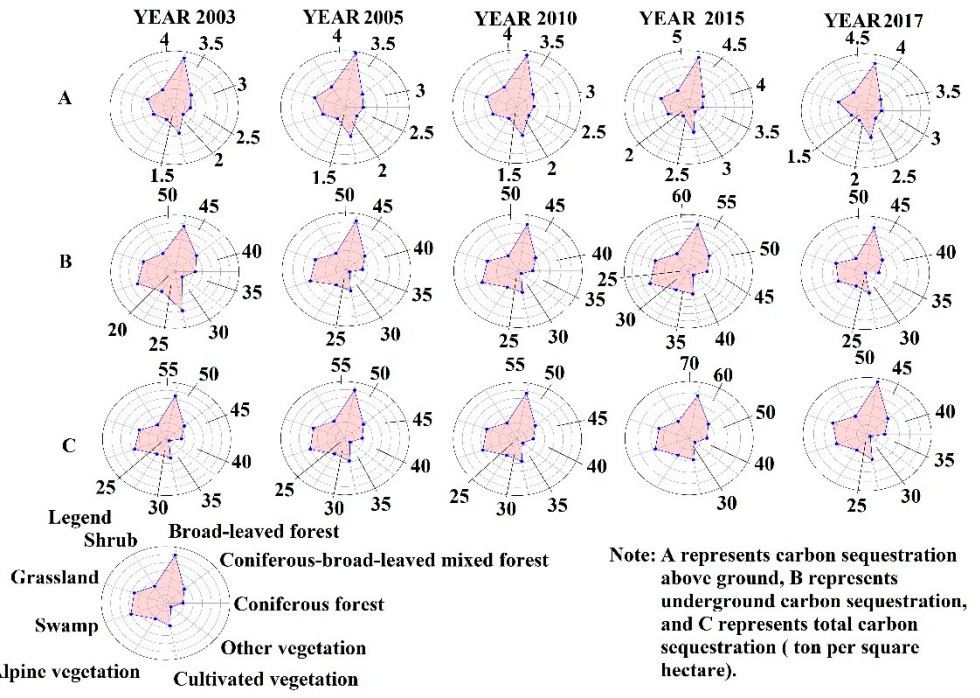

**Figure 6.** Difference of vegetation types on the above, below and total carbon sequestration.

*3.4. Estimation of Soil Organic Carbon Value under Different Scenarios*

China's carbon trading market is still in its pilot stage. Despite this, the discount rate can still directly affect the economy as it is one of the most important means of macro-control of market interest rates. Therefore, different carbon prices and discount rates were set to study the effects on carbon value. Based on past work [39], this study established six different scenarios with a mix of different discount rates (5% and 7%) and carbon prices (30 yuan/ton, 45 yuan/ton, and 60 yuan/ton) to explore the spatial pattern of soil organic carbon value in 2015 and 2017 (Figure 7). As shown, there were differences in

the value of organic carbon obtained with these different discount rates and prices. Notably, the higher the discount rate, the larger the corresponding total NPV. When the discount rate was 7% and the carbon price was 60 yuan/ton, the total carbon value was maximized.

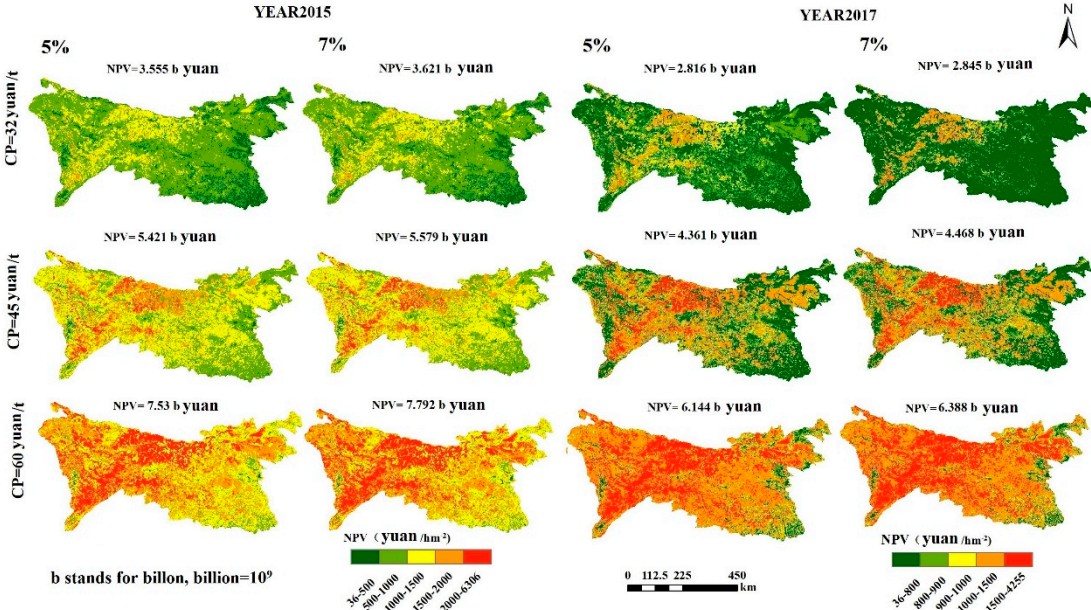

**Figure 7.** Spatial differences in soil organic carbon net present value (NPV) in 2015 and 2017 under different scenarios.

The average NPV values for each land-use type according to these different scenarios as well as the total NPV are shown in Figure 8. Different land use types based on unit of NPV and total NPV also showed differences. The order of per unit NPV was: grassland, forest land, cultivated land, and unused land; additionally, the order of total NPV was: forest land, grassland, cultivated land, and unused land. The total carbon value was lowest across the six scenarios for the unused land, and its highest value was 0.1 billion yuan (data not shown). Figure 7 also shows that the carbon price and discount rate directly impacted the total carbon price; in other words, the higher the carbon price and discount rate, the higher the NPV per unit area and the higher the total NPV for each land use type.

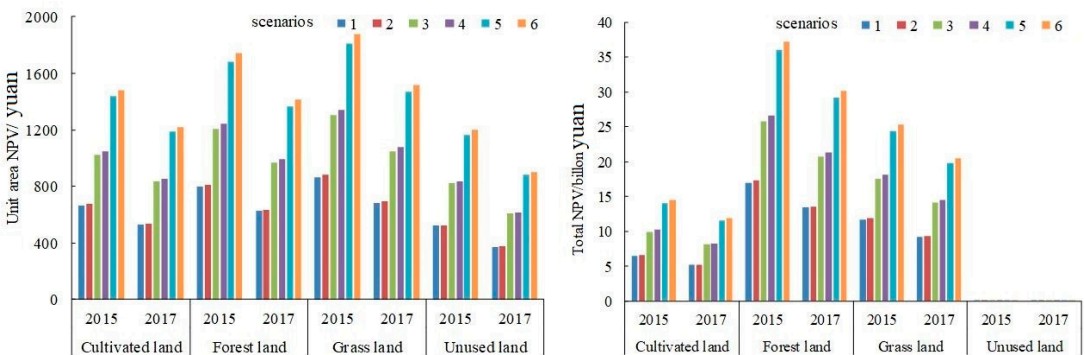

**Figure 8.** NPV mean and total value of soil organic carbon for land use under different scenarios. (Scenarios 1–6 represent each other: CP = 32 yuan t$^{-1}$, r = 5%; CP = 32 yuan t$^{-1}$, r = 7%; CP = 45 yuan t$^{-1}$, r = 5%; CP = 45 yuan t$^{-1}$, r = 7%; CP = 60 yuan t$^{-1}$, r = 5%; CP = 60 yuan t$^{-1}$, r = 7%).

## 4. Discussion

### 4.1. Soil Organic Carbon Estimation Method

Through reading the IPCC report found that the spatial and temporal variations of carbon balance on ocean warming, the arctic sea ice is melting and sea level rise have important influence, and cumulative $CO_2$ emissions on the climate of the 21st century and later had a huge impact, effective and sustainable greenhouse gas emission reduction measures is around the corner, so to improve the estimate precision of carbon, to support the economic development in the future [54].

Currently, the two main methods used in the estimation of soil organic carbon have their own advantages and disadvantages [48]. In this paper, the high temporal and spatial resolution characteristics of remote sensing were combined with the process model to reflect the dynamic changes in an ecosystem's carbon cycle. The spatial resolution of soil organic carbon was increased, and the difference between soil types was also be reflected through the spatial distribution of soil types, for example the brown earths, yellow-brown earths, skeletal soils, litho soils, paddy soils, cinnamon soils, yellow-cinnamon soils and so on.

When combining remote-sensing data, carbon cycle model, measured soil organic carbon, and meteorological observation data, the main uncertainties in estimating the spatial distribution of soil organic carbon arose from two sources [48]. First, the inconsistency in various spatial data scales was an important source, as the spatial scale of soil organic carbon measurement sample points was very small, but the spatial scale of remote sensing was relatively large [55,56]. Uncertainty caused by different scales is currently a hot research topic, but remains unresolved [57]. Second, the soil respiration was mainly affected by soil organic carbon, but also by some non-zonal factors (e.g., soil parent material, groundwater) [58]. In order to reduce the impact of these factors, improve the accuracy of soil organic carbon estimation, and reduce the impact of spatial scale inconsistency, the strategy adopted in this paper did not directly estimate soil organic carbon from either meteorological data or soil respiration data [59]. Rather, we retrieved more closely related and physically significant data relative to the soil organic carbon content; this was done by combining meteorological data, a carbon cycle process model, and remote-sensing data. More obvious indicators (i.e., soil respiration) have significantly improved the accuracy of estimating soil organic carbon.

Several researchers have estimated soil organic carbon in China using different models. Using InVEST to estimate the carbon sequestration in the Shaanxi section of the Hanjiang River; the calculation source and sink are $6.44 \times 10^4$ and $1.41 \times 10^6$ t $CO_2$, respectively, and the ecosystem service is valued by replacing the $6.07 \times 10^{11}$ yuan with the market method [60]. Carbon sequestration in Guanzhong Tianshui Economic Zone was estimated using the CASA model; the economic value of sequestered carbon was calculated using the reforestation cost method. A study shows that the carbon sequestration per unit area in 2007 was 2.821 t $CO_2$ hm$^{-2}$ a$^{-1}$, and the economic value was $2.16 \times 10^{11}$ yuan a$^{-1}$ [61]. The carbon source (mixed vegetation) and carbon footprint of Shaanxi were evaluated according to the IPCC guidelines; the carbon source level and carbon footprint in 2009 were $3.55 \times 10^9$ and $2.60 \times 10^8$ t C, respectively [62]. It can be seen from the above literature that the estimation of soil organic carbon has been relatively mature in China and is suitable for remote sensing estimation at different scales.

### 4.2. Difference of Carbon Sequestration between Aboveground and Belowground

SOC is composed of the NPP of vegetation and the intensity of soil mineralization; the intensity of soil mineralization is closely related to the regional climate conditions. Water and heat conditions affect the degree of regional soil mineralization [48]. When comparing the NPP and the spatial distribution of SOC found in the western area of this region, the NPP values were lower, but the corresponding SOC was higher. This was mainly because of the negative aspects regarding rainfall and temperature conditions [63]; in short, the decomposition rate of the soil humus lowers the amount of soil respiration [56]. Simultaneously, there is a notably amount of litter input, which ultimately promotes a higher SOC level in this region.

The main factors affecting soil organic carbon are vegetation types, climate, and human activities [39]. There are significant differences in organic carbon content and density between different vegetation types. Different vegetation types are important factors affecting SOC content and density. Vegetation types affect the concentration and distribution of SOC, this is mainly because of the different forms of vegetation on the surface, resulting in different kinds and quantities of organic matter imported into the soil. Then, there are also differences in decomposition difficulty, decomposition rate, and decomposition products. Collectively, this leads to variations in the distribution of organic carbon content. Soil organic carbon is lower in some regions as the result of myriad factors including policy, traffic, and population. Together, these factors can lead to significant changes in regional land use; this was reflected in the increased carbon reserves seen in different land-use types (i.e., cultivated land, forest land, meadow soil). Moreover, SOC decreases in locations where there is urban construction [39]. Land use changes can also change the soil structure, prompting an increase in the SOC decomposition rate. These factors can all affect a region's SOC content. At the same time, because the missing information of deep soil carbon pool and the soil thickness also varied by ecosystem type (e.g., farmland, forest, grassland), so the estimation of belowground carbon was not that even close.

The results showed that different types of land use had different carbon reserves, and forest and grassland had higher carbon reserves. But with the development of economy, more and more forests and grasslands have been transformed into urban construction land, which changes the carbon storage. Thus, land use change is often accompanied by a large amount of carbon exchange, which affects greenhouse gas emissions and land carbon storage. Therefore, land use change significantly impacts land carbon reserves. Implementing a policy of returning farmland to forestry and grassland would allow for an increased planting area for trees and grasslands, the subsequent growth of trees and the increase of defoliation will increase soil carbon content, as well as the sensitivity of soil carbon to temperature. Given the current global warming trend, a policy of returning farmland to forestry and grassland would enable plants to fix more carbon in the soil, thus promoting grassland and forests.

### 4.3. Estimation of Carbon Sequestration Value

According to the work presented here, carbon trading is a potential economic means to protect forests and yield great environmental benefits, this is because it would increase the price of trading, thereby indirectly encouraging people to protect the ecological environment [64]. Moreover, the discount rate is an estimate of future people's interests and an indicator to ensure the interests of future generations. Within a certain range, the discount rate is proportional to the value: the higher the discount rate, the greater the corresponding value, and the greater the attention paid to the interests of future generations. Given the current state of today's society, proper and reasonable increase in the discount rate and carbon price would be conducive to improving the benefits of soil organic carbon, thus promoting people's recognition of the function of carbon sequestration in the ecosystem. This would also stimulate people's protection of soil and vegetation, thus promoting greener development of the social economy; this is aligned with current trends [39].

With an increase in carbon storage, the value of carbon is higher under different carbon price and discount rate scenarios. However, this research constructed six scenarios based on different carbon prices and discount rates to explore the carbon sequestration value of soil, which is far from enough regarding the uncertainty of future developments. In future work, we hope to integrate climate change, forest age, forest management model [50,52], related policies and so on. This will allow us to build more scenarios that allow for even better simulation of real future developments.

### 4.4. Limitations

The CASA model has inherent limits as all biogeochemical models in the representation of the carbon dynamics. First, the establishment of the model is initially aimed at the vegetation in North America, and the ecological environment and climate differences around the world are relatively large, so the scale adaptability of the model needs to be studied. Second, light energy utilization is a very

important factor for accurate estimation of NPP, which is affected by soil environment, temperature, precipitation, vegetation type and other factors, among which the value of maximum light energy utilization plays a decisive role in determining the actual light energy utilization. Therefore, in the future, we should try more accurate methods to make the model more suitable for this region.

## 5. Conclusions

The high value area of soil organic carbon density is mainly distributed in the middle area of the Qinling Mountain and in the high forest coverage areas like the Daba Mountain; this is followed by the southwest area, which is covered by grasses and shrubs, and includes the Kunlun and Minshan Mountain, and areas with cultivated land distribution have low soil carbon density. From 2003 to 2017, the average soil carbon density fluctuated over a small range in the Qinling-Daba Mountain. Soil organic carbon was affected by the interaction of nature and human society. Climate factors such as precipitation and temperature have a great influence on soil organic carbon density and this effect is mediated through soil respiration. Land use types have a great impact on soil surface carbon storage: forest land has the largest carbon density and cultivated land has the largest total carbon storage; population expansion, urban construction, and land expansion have resulted in sharp decreases in the soil carbon storage of some cities (districts). Carbon prices and discount rates can be reasonably set and effectively adjusted to allow for improved soil carbon sequestration benefits. The results show that different carbon price and discount rate have significant effect on the net present value of soil carbon sequestration. Similarly, different types of land use respond differently to carbon prices. This study warns us that while developing the economy, we should also protect the ecological environment, improve the fragile ecological environment, and promote the coordinated development of economy and ecology.

**Author Contributions:** Conceptualization and methodology, Y.Y. and J.L.; software, L.Z. and C.Z.; validation, Y.Y., C.Z. and L.Z.; formal analysis and investigation, resources, data curation, writing—original draft preparation and visualization, Y.Y.; writing-review and editing, Y.Y. and J.L.; supervision, J.L.; project administration and funding acquisition, J.L. and Z.Z. All authors have read and agreed to the published version of the manuscript.

**Funding:** This research was funded by National Science and Technology Fundamental Resources Investigation "Comprehensive Scientific Investigation of China's North-South Transitional Zone", "Basic Geographic Elements and Major Resources Investigation and Mapping: 2017FY100905, The Fundamental Research Funds For the Central Universities, Shaanxi Normal University: GK201901009, Natural Science Basic Research Plan in Shaanxi Province of China: No. 2018JM4010, National Natural Science Foundation of China: No.41771198, National Natural Science Foundation of China: No.41771576, The Fundamental Research Funds For the Central Universities, Shaanxi Normal University, No. 2019TS019.

**Conflicts of Interest:** The authors declare no conflicts of interest.

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
