# Peer review of "Estimation of the Value of Ecosystem Carbon Sequestration Services under Different Scenarios in the Central China (the Qinling-Daba Mountain Area)"

_sustainability, doi:10.3390/su12010337_

Round 1
Reviewer 1 Report
Authors should put out line number so reviewers can refer the comments to a specific location. There are lots of places with bad sentences due to grammar issue and incorrect use of punctuation. But I don't want to edit it because 1. there is a lot 2. no line number for me to easily refer to
Page 1, line 5, change "..is by increasing.." to "..is to increase..", too many "by" in this sentence.
Page 1, line 6, why soil provides water? probably say nutrients? the original water source is not from soil
Second paragraph of Intro missing a problem statement, or would there be one? Same for the third paragraph, and these two paragraph should be better organized (e.g. find a common point of these papers and write a synthesis sentence ) instead of listing all these papers one by one. This is not a professional writing.
Third paragraph of Intro, line 6, bad sentence, grammar issue
Forth paragraph of Intro, first sentence, why the combination of NPV and geography should be explored? what is the advantage? the following sentences only explains the importance of using NPV, but not using both NPV and geography.
first appearance of CASA except in Abstract, please writhe out the full name?
what is soil basic respiration? I only know soil respiration.
2.1 study area: line 6, bad sentence, grammar issue and too long.
2.1 study area, second paragraph, line 4, " on the whole" what does this mean? and I don't get the point of putting that many sentences describing the weather and explaining why. would that help illustrate the key outcome of this paper?
Figure 1 is a little blur, and the three graphs on the right are hard to see.
2.2 Data sources, why these five years, and they are not evenly distributed
better to provide a link to the meteorological data
For equation (2), why the σ is 0.4? please provide the reference source. We usually use 0.48 or 0.5 for biomass carbon content. not sure about soil organic matter because once biomass touch soils, making thing complex.
2.3.2 the subheading is "below-ground carbon sequestration", why the first sentence is "above-ground..."? and what is the variable measuring the photosynthesis process?
Table 3, would three digits be necessary?
first sentence of 3.2 section, sounds like a caption for Figure 3
For discussion, were there any published papers looking at the spatial pattern of SOC in the past? to compare with the values estimated in this paper?
returning farmland to forestry was not a easy task. Government needs to weigh economic vs. environment. because farmland provides food and jobs.
Figure 7 can be improved by merge X, Y axis, and legend to minimize space.
The estimation of belowground carbon was not that even close, because the missing information of deep soil carbon pool. Authors need to mention that. the soil thickness also varied by ecosystem type (e.g. farmland, forest, grassland).
Author Response
Response to Reviewer 1 Comments
First of all, thank you very much for reviewing the paper in your busy schedule, and providing valuable comments on the paper. The comments are very good. The author also revised your valuable comments. The details are as follows (cleaner version of authors' reply available in attachment below).
Page 1, line 5, change "..is by increasing.." to "..is to increase..", too many "by" in this sentence.
I have modified it according to your comments.
I have change "..is by increasing.." to "..is to increase.."
Page 1, line 6, why soil provides water? probably say nutrients? the original water source is not from soil
I have modified it according to your comments.
Soil is the basis of human survival, providing food, nutrition and various resources (including carbon, primary minerals, secondary minerals, organic matter and microorganisms), and also an important part of the global carbon cycle.
Second paragraph of Intro missing a problem statement, or would there be one? Same for the third paragraph, and these two paragraph should be better organized (e.g. find a common point of these papers and write a synthesis sentence ) instead of listing all these papers one by one. This is not a professional writing.
I have made changes based on your comments and summarized the literature to highlight my research purposes.
Second paragraph :In general, there are some problems in the application of remote sensing estimation, field investigation and statistical data modeling to the estimation of organic carbon. With the improvement of remote sensing inversion technology, the estimation method of organic carbon is more accurate. Therefore, this paper combines the carbon cycle process model of ecosystem with the estimation model of net primary productivity to make the simulation results more ideal.
Third paragraph: The previous study found that NPV method is applied to various fields and has certain advantages in value estimation. In this paper, NPV method is combined with soil organic carbon in geography. It is found that NPV method, including carbon price and discount rate into soil carbon fixation value model, is a more practical research method compared with afforestation cost method, carbon tax law and cost avoidance method commonly used in China Research methods.
Third paragraph of Intro, line 6, bad sentence, grammar issue
I have modified it according to your comments.
In 2014, Kong and Zhang [18] used carbon tax law to estimate the functional value of wetland carbon sequestration in the reserve, providing some reference for the functional value of wetland carbon sequestration in China.
Forth paragraph of Intro, first sentence, why the combination of NPV and geography should be explored? what is the advantage? the following sentences only explains the importance of using
NPV, but not using both NPV and geography.
I have modified it according to your comments.
China's carbon trading market is still in the pilot stage, and the discount rate has a certain directivity to the economy. NPV method is an economic method combining carbon tax and discount rate, while the soil organic carbon estimation method combining ecosystem carbon cycle process model and net primary productivity estimation model is now a more accurate estimation method, combining the economic method and remote sensing method, it is a new attempt to evaluate the value of soil organic carbon.
first appearance of CASA except in Abstract, please writhe out the full name?
I have modified it according to your comments.
I have change "CASA" to " Carnegie-Ames-Stanford approach"
what is soil basic respiration? I only know soil respiration.
I have modified it according to your comments.
I checked that the relevant literature is indeed soil respiration, and I made a mistake myself. The place where the full text of the issue has been revised.
2.1 study area: line 6, bad sentence, grammar issue and too long.
I have modified it according to your comments.
The Qinling-Daba Mountain is the upstream part of the Hanjiang River, which area of the basin in this area reaches 62,000 km2.
2.1 study area, second paragraph, line 4, " on the whole" what does this mean? and I don't get the point of putting that many sentences describing the weather and explaining why. would that help illustrate the key outcome of this paper?
I have modified it according to your comments.
The content of the study area was re-described and some content was shortened.
Figure 1 is a little blur, and the three graphs on the right are hard to see.
I have modified it according to your comments.
The diagram has been modified according to your comments so that each picture is presented more clearly.
2.2 Data sources, why these five years, and they are not evenly distributed better to provide a link to the meteorological data
Originally, there was a time period of 5 years, 2000-2017, but the daily value data of 2000 meteorological stations was deficient, and there were many missing measurements. Therefore, in order to ensure the consistency of meteorological data, 2003 was used as the starting year, resulting in uneven distribution of time periods. The data from http://data.cma.cn/site/index.html.
For equation (2), why the σ is 0.4? please provide the reference source. We usually use 0.48 or 0.5 for biomass carbon content. not sure about soil organic matter because once biomass touch soils, making thing complex.
The determination of the σ value is based on the relevant literature[38], and you said that 0.48 or 0.5 is also mentioned in the relevant literature. According to what you said, there is a certain uncertainty in the soil organic matter, because once the biomass comes into contact with the soil, it will make There are certain reasons for complicating things.
2.3.2 the subheading is "below-ground carbon sequestration", why the first sentence is "above-ground..."? and what is the variable measuring the photosynthesis process?
Underground carbon sequestration, also known as soil carbon pool, is estimated on the basis of the amount of carbon sequestration on the ground, and the underground carbon storage capacity of different land use types is different, so the underground carbon sequestration is estimated according to a certain proportion.
Photosynthesis process is because the CASA model used in estimating NPP involves factors such as solar radiation, temperature, and precipitation, so the npp value is estimated under solar photosynthesis.
Table 3, would three digits be necessary?
Listen to your comments and keep the numbers in the text to a decimal.
first sentence of 3.2 section, sounds like a caption for Figure 3
The first sentence is an expanded description of Figure 3, so the feeling is a bit like the name; Figure 3. I have listened to your suggestion and modified the description in the text.
For discussion, were there any published papers looking at the spatial pattern of SOC in the past? to compare with the values estimated in this paper?
The spatial and temporal changes of soil organic carbon in Qinling-Daba mountains are rarely studied. Most of them are based on the model to simulate the spatiotemporal variation of npp. Soil organic carbon is more studied in the other area and can be used as a reference.
Several researchers have estimated soil organic carbon in China using different models. Using InVEST to estimate the carbon sequestration in the Shaanxi section of the Hanjiang River; the calculation source and sink are 6.44×104 and 1.41×106 t CO2, respectively, and the ecosystem service is valued by replacing the 6.07×1.011 RMB with the market method [54]. Carbon sequestration in Guanzhong Tianshui Economic Zone was estimated using the CASA model; the economic value of sequestered carbon was calculated using the reforestation cost method. A study shows that the carbon sequestration per unit area in 2007 was 2.821 t CO2 hm-2 a-1, and the economic value was 2.16×1011RMB a-1 [55]. The carbon source (mixed vegetation) and carbon footprint of Shaanxi were evaluated according to the IPCC guidelines; the carbon source level and carbon footprint in 2009 were 3.55×109 and 2.60×108 t C, respectively[56]. It can be seen from the above literature that the estimation of soil organic carbon has been relatively mature in China and is suitable for remote sensing estimation at different scales.
returning farmland to forestry was not a easy task. Government needs to weigh economic vs. environment. because farmland provides food and jobs.
Returning farmland to forests is indeed a big ecological project. The government has to weigh all aspects, including the harmonious development of the economy and the ecological
environment. The implementation of the policy of returning farmland to forests and grassland has indeed played a positive role in ecological environmental protection, reducing soil erosion, mudslides, sandstorms and other natural disasters, while improving the environmental quality of some areas and promoting the coordinated development of economy and environment.
Figure 7 can be improved by merge X, Y axis, and legend to minimize space.
According to your requirements, Figure 7 has been modified to make the space more reasonable.
The estimation of belowground carbon was not that even close, because the missing information of deep soil carbon pool. Authors need to mention that. the soil thickness also varied by ecosystem type (e.g. farmland, forest, grassland).
In accordance with your request, this article has been revised to increase the problems of this paper and the inadequacies of the research methods.
At the same time, because the missing information of deep soil carbon pool and the soil thickness also varied by ecosystem type (e.g. farmland, forest, grassland), so the estimation of belowground carbon was not that even close.

Reviewer 2 Report
sustainability-625653
The manuscript entitled “Estimation of the value of ecosystem carbon sequestration services under different scenarios in the central China (the Qinling-Daba Mountain area)” by Yu et al., dealing with an interesting topic to quantify soil carbon sequestration. There is no line number for MS and it is difficult to address the comment. However, the comments provided based on the page (P).
General comment:
The title can be more appealing.
Abstract
Please add some numerical results to the Abstract to support your statements.
P1: change “Key words” to “Keywords” without space.
P1: Do not use the abbreviations for Keywords.
Introduction
P1: rewrite this sentence “Soil provides human beings with food, water, and various resources and forms the basis for human survival”. It is not clear. What type of “various resources” and “forms the basis”.
P1: Change “SOC decomposition” to “soil organic carbon decomposition”. For the first time define the abbreviation.
P2: recently some machine learning methods frequently used to predict SOC. Their requirements are similar to the Geostatistical methods but with robust prediction. So, you can add some recent studies of Machine learning to the following sentence to cover wider topics. “Geostatistical methods [4] are commonly used to study the spatial variability of SOC, and include field surveys and statistical data modeling methods [5-6].” The following works are useful:
Hinge, G., Surampalli, R.Y., Goyal, M.K., 2018. Prediction of soil organic carbon stock using digital mapping approach in humid India. Environmental earth sciences 77(5), 172.
Zeraatpisheh, M., Ayoubi, S., Jafari, A., Tajik, S., Finke, P., 2019. Digital mapping of soil properties using multiple machine learning in a semi-arid region, central Iran. Geoderma 338, 445-452.
Ratnayake, R., Karunaratne, S., Lessels, J., Yogenthiran, N., Rajapaksha, R., Gnanavelrajah, N., 2016. Digital soil mapping of organic carbon concentration in paddy growing soils of Northern Sri Lanka. Geoderma Regional 7(2), 167-176.
P2: NPP estimation model and achieved ideal estimation results[14]. Define “NPP”.
P2: long sentence. It would be better split into two sentences. “In 2011, Li et al [15] estimated the distribution characteristics and variations in carbon sequestration and oxygen release values in the Loess Plateau of Northern Shaanxi, China, they used two methods: afforestation cost and industrial oxygen production.”
P2: I am not sure this type of referring to previous studies without mention to the citation is correct or not. In 2014, Kong applied the carbon tax method to study Zhang Wei and evaluated the wetland carbon sequestration in the Heihe protected area[16].” Since Kong and Zhang Wei are no recognized methods, so it would be better to add the citation exactly after the names.
P2: The NPV method refers to the difference between. Define “NPV”.
P2: Finally, Nghiem et al [26]. Why do you use finally? The reader probably misunderstood that this sentence is the conclusion of the previous sentences.
P2: Nghiem et al [26] used this same method. Which “this same method”, “this same method” refers to what?
P3: of soil basic respiration using a CASA model. Define “CASA”.
P3: at the end of the introduction the aims of the study should rewrite. In the current format, it is difficult to follow. Probably the numeric aim is a good option (i: , ii: , iii: ,….).
Study area and data
Well organized.
P4: “soil resources and various types, mainly including brown earths, yellow-brown earths, skeletol soils, litho soils, paddy soils , cinnamon soils , yellow-cinnamon soils and so on[35].” I think these soil names or classifications are not clear for international readers, I recommend you provide WRB or Soil Taxonomy classifications.
P5: change “normalized vegetation index (NDVI)” to “normalized difference vegetation index (NDVI”.
Result
Well written and structure.
Figure 3. the caption is not complete and independent.
Discussion
The discussion needs more supportive references. Mainly sections 4.2 and 4.3 do not have references. How the readers can be assured about these results and discussion while there are no supporting studies or logical references?
The discussion section did not follow the Results section. It should discuss the topics reported in the Result section.
P14:”4.1 Uncertainty of soil organic carbon estimation method” In the materials $ Method section there is no methodology for assessing Uncertainty. Also in the results section, there are no results regarding Uncertainty
Conclusion
P 15: “To this end, the higher the carbon price and discount rate, the higher the carbon sequestration income of the soil. The NPV per unit area of woodland was the highest; contrastingly, the NPV per unit area
of land used and cultivated land was low.” Please rewrite this sentence.
Author Response
Response to Reviewer 2 Comments
First of all, thank you very much for reviewing the paper in your busy schedule, and providing valuable comments on the paper. The comments are very good. The author also revised your valuable comments. The details are as follows. (cleaner version of authors' reply available in attachment below).
Abstract
Please add some numerical results to the Abstract to support your statements.
P1: change “Key words” to “Keywords” without space.
P1: Do not use the abbreviations for Keywords.
Add a digital statement to the abstract to support the paper according to your opinion
The net present value of soil carbon sequestration in the six scenarios in 2015 was 35.55×108 RMB, 36.209×108 RMB, 54.213×108 RMB, 55.793×108 RMB, 75.3×108 RMB, 77.929×108 RMB; The net present value of soil carbon sequestration in 6 scenarios in 2017 is 28.16×108 RMB, 28.451×108 RMB, 43.610×108 RMB, 44.667×108 RMB, 61.44×108 RMB, 63.38×108 RMB.
I have changed "Key words" to "Keywords"
I have changed "NPP" to " net primary productivity " , "SOC" to " soil organic carbon ", "NPV" to " net present value ".
Introduction
P1: rewrite this sentence “Soil provides human beings with food, water, and various resources and forms the basis for human survival”. It is not clear. What type of “various resources” and “forms the basis”.
Soil is the basis of human survival, providing food, nutrition and various resources (including carbon, primary minerals, secondary minerals, organic matter and microorganisms), and also an important part of the global carbon cycle
P1: Change “SOC decomposition” to “soil organic carbon decomposition”. For the first time define the abbreviation.
I have changed " SOC decomposition " to " soil organic carbon decomposition " .
P2: recently some machine learning methods frequently used to predict SOC. Their requirements are similar to the Geostatistical methods but with robust prediction. So, you can add some recent studies of Machine learning to the following sentence to cover wider topics. “Geostatistical methods [4] are commonly used to study the spatial variability of SOC, and include field surveys and statistical data modeling methods [5-6].” The following works are useful:
Thank you very much for your comments. It is very good to apply the machine learning method to the estimation of soc, and it also has a good effect. This article uses several of the references you have suggested and summarizes and summarizes them.
Geostatistical methods [4-6] are commonly used to study the spatial variability of SOC, and recently some machine learning methods frequently used to predict SOC[7-9]. Many non-linear models including Cubist (Cu), Random Forest (RF), Regression Tree (RT) and a Multiple Linear Regression (MLR) were used to simulate the distribution of soil carbon reserves, in order to predict and generate continuous spatial clear soil carbon map[7-9].
P2: NPP estimation model and achieved ideal estimation results[14]. Define “NPP”.
NPP refers to the amount of organic matter accumulated by green plants in unit time and unit area, which is the remaining part of the total amount of organic matter produced by plant photosynthesis minus autotrophic respiration).
P2: long sentence. It would be better split into two sentences. “In 2011, Li et al [15] estimated the distribution characteristics and variations in carbon sequestration and oxygen release values in the Loess Plateau of Northern Shaanxi, China, they used two methods: afforestation cost and industrial oxygen production.”
In 2011, Li and Ren [17] estimated the distribution characteristics and variations in carbon sequestration and oxygen release values in the Loess Plateau of Northern Shaanxi, China, they used two methods: afforestation cost and industrial oxygen production.
P2: I am not sure this type of referring to previous studies without mention to the citation is correct or not. In 2014, Kong applied the carbon tax method to study Zhang Wei and evaluated the wetland carbon sequestration in the Heihe protected area[16].” Since Kong and Zhang Wei are no recognized methods, so it would be better to add the citation exactly after the names.
In 2014, Kong and Zhang [18] used carbon tax law to estimate the functional value of wetland carbon sequestration in the reserve, providing some reference for the functional value of wetland carbon sequestration in China. In 2014, Pang et al. [19] used the market value method to quantitatively evaluate the ecosystem service value provided by the Zoige alpine wetlands.
P2: The NPV method refers to the difference between. Define “NPV”.
The net present value of a project is the present value of current and future benefit minus the present value of current and future costs.
P2: Finally, Nghiem et al [26]. Why do you use finally? The reader probably misunderstood that this sentence is the conclusion of the previous sentences.
The site was modified according to your comments.
Nghiem et al. [28] used NPV method to determine how to estimate the conservation benefits of carbon sequestration and biodiversity under different forest management models to maximize the NPV of timber sales.
P2: Nghiem et al [26] used this same method. Which “this same method”, “this same method” refers to what?
This same method” refers to the NPV method.
P3: of soil basic respiration using a CASA model. Define “CASA”.
The CASA model mainly extracts normalized difference vegetation index (NDVI) from remote sensing images; the model was used to estimate the NPP of the terrestrial global ecosystem using the principle of light energy utilization.
P3: at the end of the introduction the aims of the study should rewrite. In the current format, it is difficult to follow. Probably the numeric aim is a good option (i: , ii: , iii: ,….).
According to your comments, this article summarizes each paragraph of the introduction and highlights the research purpose of the article, so that the research has certain scientific significance.
Study area and data
Well organized.
P4: “soil resources and various types, mainly including brown earths, yellow-brown earths, skeletol soils, litho soils, paddy soils , cinnamon soils , yellow-cinnamon soils and so on[35].” I think these soil names or classifications are not clear for international readers, I recommend you provide WRB or Soil Taxonomy classifications.
I have change “normalized vegetation index (NDVI)” to “normalized difference vegetation index (NDVI”.
According to your opinion, the description of the soil classification system has been added, see Table 2.
Table 2 Classification system of soil types(The Classification system of soil types comes from the Resource and Environment Data Center of Chinese Academy of Sciences-Resource and Environment Data Cloud Platform. http://www.resdc.cn/DataSearch.aspx)
|
Soil Order |
Great Soil Group |
|
Leached soil |
Brown coniferous forest soils |
|
Brown earths |
|
|
Yellow-brown earths |
|
|
Yellow-cinnamon soils |
|
|
Brown earths |
|
|
Dark-brown earths |
|
|
Albic soils |
|
|
Semi-leaching soil |
Torrid red soils |
|
Cinnamon soils |
|
|
Grey-cinnamon soils |
|
|
Black soils |
|
|
Grey forest soils |
|
|
Calcium layer soil |
Chernozems |
|
Castanozems |
|
|
Castano-cinnamon soils |
|
|
Dark loessial soils |
|
|
Arid soil |
Brown pedocals |
|
Sierozems |
|
|
Desert soil |
Gray desery soils |
|
Gray-brown desrt soils |
|
|
Brown desert soils |
|
|
Primary soil |
Cultivated loessial soils |
|
Red clay soils |
|
|
Alluvial soils |
|
|
Takyr soils |
|
|
Aeolian soils |
|
|
Limestone soils |
|
|
Volcanic soils |
|
|
Purplish soils |
|
|
Litho soils |
|
|
Skeletol soils |
|
|
Semi-aqueous soil |
Meadow soils |
|
Lime concretion black soils |
|
|
Mountain meadow soils |
|
|
Shruby meadow soils |
|
|
Fluvo-aquic soils |
|
|
Aqueous soil |
Bog soils |
|
Peat soils |
|
|
Saline soil |
Saline soils |
|
Desert solonchaks |
|
|
Coastal solonchaks |
|
|
Acid sulphate soils |
|
|
Frigid plateau solonchaks |
|
|
Solonetzs |
|
|
Artificial soil |
Paddy soils |
|
Cumulated irrigated soils |
|
|
Irrigated desert soils |
|
|
High-Mountain-Soils |
Felty soils |
|
Dark felty soils |
|
|
Frigid calcic soils |
|
|
Cold calcic soils |
|
|
Cold brown calcic soils |
|
|
Frigid desert soils |
|
|
Cold desert soils |
|
|
Frigid frozen soils |
|
|
Iron bauxite |
Humid-thermo ferralitic |
|
Lateritic red earths |
|
|
Red earths |
|
|
Yellow earths |
Result
Figure 3. the caption is not complete and independent.
The title of Figure 3 is supplemented by your comments.
Figure 3. Spatial and temporal distribution of NPP、soil respiration、soil organic carbon in 2003-2017
Discussion
The discussion needs more supportive references. Mainly sections 4.2 and 4.3 do not have references. How the readers can be assured about these results and discussion while there are no supporting studies or logical references?
According to your comments, the literature references in the discussion section have been added to make the article more convincing.
The discussion section did not follow the Results section. It should discuss the topics reported in the Result section.
The discussion 4.1 is based on the selection of organic carbon assessment methods, mainly based on the combination of remote sensing inversion and carbon cycle process model, while most of the research is based on field detection and remote sensing image direct estimation methods, each method has certain defects, so this paper is to combine the two methods; 4.2 is to discuss On the reasons of influencing the change of carbon sequestration, the paper discusses the influence of natural factors and the change of land use on the carbon sequestration. 4.3 based on the estimation of carbon sequestration value, the paper analyzes the space-time change of carbon sequestration value according to the scenario setting of different carbon price and discount rate, which is more suitable to the reality. At the same time, there are some problems in this method For example, the value of carbon sequestration is related to many factors. This paper only deals with the discount rate and carbon price. The scenario involves too few aspects. In the future, we need to further study this aspect.
P14:”4.1 Uncertainty of soil organic carbon estimation method”In the materials Method section there is no methodology for assessing Uncertainty. Also in the results section, there are no results regarding Uncertainty
This section discusses the choices mainly related to the method of estimating carbon sequestration, so the title is somewhat ambiguous and has been revised in this paper.
Conclusion
P 15: “To this end, the higher the carbon price and discount rate, the higher the carbon sequestration income of the soil. The NPV per unit area of woodland was the highest; contrastingly, the NPV per unit area of land used and cultivated land was low.” Please rewrite this sentence.
The conclusions of the article have been reorganized and expressed according to your opinions.
Studies have shown that different carbon prices and discount rates have significant effects on NPV in soil carbon sequestration. Similarly, different land use types have different responses to total carbon prices.

Reviewer 3 Report
Sustainability-625653 presents an interesting study aiming to quantify above and below-ground carbon sinks using the Carnegie-Ames-Stanford approach (CASA) model and an improved carbon cycle process model in Qinling-Daba Mountain. Further this study explores the potential of NPV for carbon value estimation in different scenarios. The manuscript is well written and well estructured. Please see below some minor comments:
Introduction
“in the atmosphere” instead of “in the global atmosphere”
Include a reference “rising global temperatures”
This sentence “Soil provides human beings with food, water, and various resources and forms the basis for human sur vival.” Could be ommitted
Zhou et al. combined … results Where?
Please revise the sentence “In 2014, Kong applied the carbon tax method to study Zhang Wei and evaluated the wetland car bon sequestration”
“The Net Present Value (NPV) method referes …”
This sentence seems incomplete perfect for? “because its theoretical basis is relatively perfect”
Please include some references and explain in more detail “The value of the carbon sequestration service is more realistic than the domestically accepted method; moreover, the method used to estimate the effect is more robust.”
to estimate WHAT? in Qinling Daba Mountain
Study area and dataTHis sentence can be ommited “The Qinling-Daba Mountain area is rich in water, forest, and grass resources.” Or compared with other parts of China top ut in value the richness of the area
the Qinling-Daba Mountain affects the cold air
Figure should be improved to show in a bigger size legends and maps
A figure including the methodology scheme could be very illustrative
Table 1 some of the definitions are obvious for example “Shrubs are vegetation types dominated by shrubs” Authors can consider to replace as follows “Shrubs are vegetation types dominated by – include here the vegetation speces – Similarly in Broad-leaf forest
Other vegetation Unused land – Please specify which other species
2.3.1 Aboveground carbon sequestration
Please define the Carnegie-Ames-Stanford model (CASA) mainly …
Add a reference for the following sentence (e.g. Monteith 1972) NPP of the vegetation using two parameters: the absorbed photosynthetic active radiation(APAR)and the light energy conversion rate
2.3.2 Below-ground carbon sequestration
Move “process. Remote-sensing images described the differences between cultivatedland, forest, and grassland. “ after “different communities (i.e., cultivated land, forests,and grasslands)”
Include a reference for this approach “. This approach accounted for the differences between litter and root biomass in different communities.”
Zhou et al. published an approach to improve the carbon cycle
Equation 5: Define X
ResultsMove to materials and methods: “Based on the remote sensing data, the CASA model was used to estimate the mass of NPP across 2003-2017.”
Lower case “Broad-leavedforest,Cultivated vegetation,Grassland,Shrub,Swamp,Coniferous-broad-leaved mixed forest,Coniferous forest, Alpine vegetation, and Other vegetation”
Table 3 remove one bracket in year
correspondingly lower ability to decrease vegetation cover in the mountainous
Move to Materials & Methods “the annual spatial and temporal distribution of soil basic respiration was obtained by combining the NPP data obtainedfrom the CASA model with the carbon cycle process model.”
Move to conclusions “ecological environment, making carbon storage decline. This study warns us that while developing economy, we should also protect the ecological environment, improve the fragile ecological environment and promote the coordinated development of economy and ecology.”
Please revise, this paragraph is not a result from this study “The content of soil organic carbon depends on the input and output of soil organic carbon; that is, the content of soil organic carbon is determined by the net primary productivity of vegetation and the mineralization intensity of soil organic carbon. Mineralization intensity, in turn, is mainly affected by hydrothermal conditions.”
In the discussion Authors referred to “Land use changes are often accompanied by a large amount of carbon exchange,” However it is not clear how the approaches used takes into account these changes?
Author Response
Response to Reviewer 3 Comments
First of all, thank you very much for reviewing the paper in your busy schedule, and providing valuable comments on the paper. The comments are very good. The author also revised your valuable comments. The details are as follows.
Sustainability-625653 presents an interesting study aiming to quantify above and below-ground carbon sinks using the Carnegie-Ames-Stanford approach (CASA) model and an improved carbon cycle process model in Qinling-Daba Mountain. Further this study explores the potential of NPV for carbon value estimation in different scenarios. The manuscript is well written and well estructured. Please see below some minor comments:
Introduction
“in the atmosphere” instead of “in the global atmosphere”
Thank you for your suggestion. It has been revised as you said.
Include a reference “rising global temperatures”
Thank you for your suggestion. It has been revised as you said.
This sentence “Soil provides human beings with food, water, and various resources and forms the basis for human survival.” Could be ommitted
Soil is the basis of human survival, providing food, nutrition and various resources (including carbon, primary minerals, secondary minerals, organic matter and microorganisms), and also an important part of the global carbon cycle.
Zhou et al. combined … results Where?
Due to the negligence of manuscript revision, this reference is a bit inappropriate in this place, so the author transferred this reference to other places, which is more reasonable.
Please revise the sentence “In 2014, Kong applied the carbon tax method to study Zhang Wei and evaluated the wetland car bon sequestration”
In 2014, Kong and Zhang [19] used carbon tax law to estimate the functional value of wetland carbon sequestration in the reserve, providing some reference for the functional value of wetland carbon sequestration in China.
“The Net Present Value (NPV) method referes …”
This sentence seems incomplete perfect for? “because its theoretical basis is relatively perfect”
NPV(Net present value)primarily measures the profit indicators of economic evaluation; it has important economic implications and is widely used across fields because its theoretical basis is relatively perfect.
Please include some references and explain in more detail “The value of the carbon sequestration service is more realistic than the domestically accepted method; moreover, the method used to estimate the effect is more robust.”
In the introduction, traditional methods of calculating carbon sequestration value are introduced, such as
In 2011, Li and Ren [18] estimated the distribution characteristics and variations in carbon sequestration and oxygen release values in the Loess Plateau of Northern Shaanxi, China, they used two methods: afforestation cost and industrial oxygen production. In 2014, Kong and Zhang [19] used carbon tax law to estimate the functional value of wetland carbon sequestration in the reserve, providing some reference for the functional value of wetland carbon sequestration in China. In 2014, Pang et al. [20] used the market value method to quantitatively evaluate the ecosystem service value provided by the Zoige alpine wetlands.
China's carbon trading market is still in the pilot stage, and the discount rate has a certain directivity to the economy. NPV method is an economic method combining carbon tax and discount rate, while the soil organic carbon estimation method combining ecosystem carbon cycle process model and net primary productivity estimation model is now a more accurate estimation method, combining the economic method and remote sensing method, it is a new attempt to evaluate the value of soil organic carbon.
to estimate WHAT? in Qinling Daba Mountain
What refers to the use of NPV method to evaluate the value of carbon sequestration in Qinling-Daba mountain area.
Study area and data
THis sentence can be ommited “The Qinling-Daba Mountain area is rich in water, forest, and grass resources.” Or compared with other parts of China top ut in value the richness of the area
It is more reasonable for the author to delete this sentence according to your suggestion.
the Qinling-Daba Mountain affects the cold air
As the reviewers suggest that the climate in this study area should be deleted, because the climate has little impact on this paper, which is not the key part of this article, so the author has simplified the description of climate, making the article more compact and reasonable.
Figure should be improved to show in a bigger size legends and maps
Thank you for your suggestion. It has been revised as you said.
A figure including the methodology scheme could be very illustrative
Thank you for your suggestion. It has been revised as you said.
Fig. 2 Framework of study
Table 1 some of the definitions are obvious for example “Shrubs are vegetation types dominated by shrubs” Authors can consider to replace as follows “Shrubs are vegetation types dominated by – include here the vegetation speces – Similarly in Broad-leaf forest
Thank you for your suggestion. It has been revised as you said.
Broad-leaf forest is composed of broad-leaved trees, with broad leaves, compared with coniferous forest and common leaves.
Shrubs are vegetation types dominated by include here the vegetation speces.
Other vegetation Unused land – Please specify which other species
Thank you for your suggestion. It has been revised as you said.
Other vegetation types include water area, unused land, wasteland, etc.
2.3.1 Aboveground carbon sequestration
Please define the Carnegie-Ames-Stanford model (CASA) mainly …
CASA model is a kind of model to estimate NPP.
The CASA [37-38] model mainly extracts normalized difference vegetation index (NDVI) from remote sensing images; the model was used to estimate the NPP of the terrestrial global ecosystem using the principle of light energy utilization.
Add a reference for the following sentence (e.g. Monteith 1972) NPP of the vegetation using two parameters: the absorbed photosynthetic active radiation(APAR)and the light energy conversion rate
Thank you for your suggestion. I have revised it according to what you said and added relevant references.
2.3.2 Below-ground carbon sequestration
Move “process. Remote-sensing images described the differences between cultivatedland, forest, and grassland. “ after “different communities (i.e., cultivated land, forests,and grasslands)”
Thank you for your suggestion. I have revised it according to what you said
Include a reference for this approach “. This approach accounted for the differences between litter and root biomass in different communities.”
Thank you for your suggestion. I have revised it according to what you said and added relevant references.
Zhou et al. published an approach to improve the carbon cycle
Thank you for your suggestion. It has been revised as you said.
Equation 5: Define X
Thank you for your suggestion. It has been revised as you said.
x is the ratio of precipitation to evapotranspiration.
Results
Move to materials and methods: “Based on the remote sensing data, the CASA model was used to estimate the mass of NPP across 2003-2017.”
Thank you for your suggestion. It has been revised as you said.
Lower case “Broad-leavedforest,Cultivated vegetation,Grassland,Shrub,Swamp,Coniferous-broad-leaved mixed forest,Coniferous forest, Alpine vegetation, and Other vegetation”
Thank you for your suggestion. It has been revised as you said.
Table 3 remove one bracket in year
Thank you for your suggestion. It has been revised as you said.
correspondingly lower ability to decrease vegetation cover in the mountainous
Thank you for your suggestion. It has been revised as you said.
Move to Materials & Methods “the annual spatial and temporal distribution of soil basic respiration was obtained by combining the NPP data obtainedfrom the CASA model with the carbon cycle process model.”
Thank you for your suggestion. It has been revised as you said.
Move to conclusions “ecological environment, making carbon storage decline. This study warns us that while developing economy, we should also protect the ecological environment, improve the fragile ecological environment and promote the coordinated development of economy and ecology.”
Thank you for your suggestion. It has been revised as you said.
Please revise, this paragraph is not a result from this study “The content of soil organic carbon depends on the input and output of soil organic carbon; that is, the content of soil organic carbon is determined by the net primary productivity of vegetation and the mineralization intensity of soil organic carbon. Mineralization intensity, in turn, is mainly affected by hydrothermal conditions.”
It mainly involves the composition of soil organic carbon. In different mountain areas, the composition of organic carbon is changing, which leads to the spatial change of organic carbon reserves in mountain areas. This point of view is obtained by referring to the research results of others, so references are added in this paper.
In the discussion Authors referred to “Land use changes are often accompanied by a large amount of carbon exchange,” However it is not clear how the approaches used takes into account these changes?
According to the results of this study, it is found that there are great differences in the carbon reserves of different land use types, and the carbon reserves of forest and grassland are higher. But with the development of economy, more and more forests and grasslands are converted into urban construction land, which makes the carbon storage change. Thus we can see land use changes are often accompanied by a large amount of carbon exchange, affecting greenhouse gas emissions and land carbon reserves.

Round 2
Reviewer 1 Report
I find this paper can be accepted as it is.
Author Response
Thank you very much for reviewing the paper in your busy schedule, and providing valuable comments on the paper. The comments are very good. The author also revised your valuable comments. According to your comments, the department has been modified.

Reviewer 2 Report
Second revision for manuscript sustainability-625653
The authors carefully address all the comments raised during the first round of the revision. However, still there is a correction or modification listed below:
P1, L23-26: to avoid repeating 108 you can increase the number of decimal then use (billion=109; b). for example, instead of 35.55×108 RMB you can write 3.555 b RBM.
Author Response
First of all, thank you very much for reviewing the paper in your busy schedule, and providing valuable comments on the paper. The comments are very good. The author also revised your valuable comments. The details are as follows.
P1, L23-26: to avoid repeating 108 you can increase the number of decimal then use (billion=109; b). for example, instead of 35.55×108 RMB you can write 3.555 b RBM.
According to your comments, the department has been modified.
